# A Proximal Algorithm for Sampling

**Jiaming Liang**                                            *jiaming.liang@yale.edu*
*Department of Computer Science, Yale University, New Haven, CT 06511.*
**Yongxin Chen**                                             *yongchen@gatech.edu*
*School of Aerospace Engineering, Georgia Institute of Technology, Atlanta, GA 30332.*

**Reviewed on OpenReview:** *https: // openreview. net/ forum? id= CkXOwlhf27*

## Abstract

We study sampling problems associated with potentials that may lack smoothness or convexity. Departing from the standard smooth setting, the potentials are only assumed to be weakly smooth or non-smooth, or the summation of multiple such functions. We develop a sampling algorithm that resembles proximal algorithms in optimization for this challenging sampling task. Our algorithm is based on a special case of Gibbs sampling known as the alternating sampling framework (ASF). The key contribution of this work is a practical realization of the ASF based on rejection sampling for both non-convex and convex potentials that are not necessarily smooth. In almost all the cases of sampling considered in this work, our proximal sampling algorithm achieves a better complexity than all existing methods.

## 1    Introduction

The problem of drawing samples from an unnormalized probability distribution plays an essential role in data science and scientific computing (Durmus et al., 2018; Clarage et al., 1995; Maximova et al., 2016). It has been widely used in many areas such as Bayesian inference, Bayesian neural networks, probabilistic graphical models, biology, and machine learning (Gelman et al., 2013; Kononenko, 1989; Koller & Friedman, 2009; Krauth, 2006; Sites Jr & Marshall, 2003; Durmus et al., 2018). Compared with optimization oriented methods, sampling has the advantage of being able to quantify the uncertainty and confidence level of the solution, and often provides more reliable solutions to engineering problems. This advantage comes at the cost of higher computational cost. It is thus important to develop more efficient sampling algorithms.

In the classical setting of sampling, the potential function $f$ of an unnormalized target distribution $\exp(-f(x))$ is assumed to be smooth and (strongly) convex. Over the past decades, many sampling algorithms have been developed, including Langevin Monte Carlo (LMC), kinetic Langevin Monte Carlo (KLMC), Hamiltonian Monte Carlo (HMC), Metropolis-adjusted Langevin algorithm (MALA), etc (Dalalyan, 2017; Grenander & Miller, 1994; Parisi, 1981; Roberts & Tweedie, 1996; Dalalyan & Riou-Durand, 2020; Bou-Rabee & Hairer, 2013; Roberts & Stramer, 2002; Roberts & Tweedie, 1996; Neal, 2011). Many of these algorithms are based on some type of discretization of the Langevin diffusion or the underdamped Langevin diffusion. They resemble the gradient-based algorithms in optimization. These algorithms work well for (strongly) convex and smooth potentials; many non-asymptotic complexity bounds have been proven. However, the cases where either the convexity or the smoothness is lacking are much less understood (Chewi et al., 2021; Chen et al., 2022; Chatterji et al., 2020; Erdogdu & Hosseinzadeh, 2021; Liang & Chen, 2022; Dalalyan et al., 2022; Balasubramanian et al., 2022; Mou et al., 2022a).

In this paper, we consider the challenging task of sampling from potentials that are not smooth and even not convex. Many sampling problems in real applications fall into this setting. For instance, Bayesian neural networks are highly non-convex models corresponding to probability densities with multi-modality (Izmailov et al., 2021). The lack of smoothness is due to the use of activation functions such as ReLU. The goal of this work is to develop an efficient algorithm with provable guarantees for a class of potentials that are not smooth. In particular, we consider potential functions that are semi-smooth, defined by (1).

Inspired by the recent line of research that lies in the interface of sampling and optimization, we examine this sampling task from an optimization perspective. We build on the intuition of proximal algorithms for non-smooth optimization problems and develop a proximal algorithm to sample from non-convex and semi-smooth potentials. Our algorithm is based on the alternating sampling framework (ASF) (Lee et al., 2021) developed recently to sample from strongly convex potentials. In a nutshell, the ASF is a Gibbs sampler over a carefully designed augmented distribution of the target and one can thus sample from the target distribution by sampling from the augmented distribution. The convergence results of ASF have been recently improved (Chen et al., 2022) to cover non-log-concave distributions that satisfy functional inequalities such as the Logarithmic Sobolev inequality (LSI) and the Poincaré inequality (PI).

The ASF is an idealized algorithm that is not directly implementable. In each iteration, it needs to query the so-called restricted Gaussian oracle (RGO), which is itself a sampling task from a quadratically regularized distribution $\exp(-f(x) - \frac{1}{2\eta}\|x-y\|^2)$ for some given $\eta > 0$ and $y \in \mathbb{R}^d$. The RGO can be viewed as a sampling counterpart of the proximal map in optimization. The total complexity of ASF thus depends on that of the RGO. Except for a few special cases where $f$ has certain structures, the RGO is usually a challenging task. One key contribution of this work is a practical and efficient algorithm for RGO for potentials that are neither smooth nor convex. This algorithm extends the recent work Liang & Chen (2022) for convex and non-smooth potentials. Combining the ASF and our algorithm for RGO, we establish a proximal algorithm for sampling from (convex or non-convex) semi-smooth potentials. Note that the techniques used in Liang & Chen (2022) are no longer applicable to these non-convex semi-smooth settings. In this work, we developed a new algorithm and associated proof techniques for efficient RGO implementation.

Our contributions are summarized as follows. i) We develop an efficient sampling scheme of RGO for semi-smooth potentials which can be either convex or non-convex and bound its complexity with a novel technique. ii) We combine our RGO scheme and the ASF to develop a sampling algorithm that can sample from both convex and non-convex semi-smooth potentials. Our algorithm has a better non-asymptotic complexity than almost all existing methods in the same setting. iii) We further extend our algorithms for potentials that are the summation of multiple semi-smooth functions.

Table 1: Complexity bounds for sampling from non-convex potentials.

| Source | Complexity | Assumption | Metric |
|---|---|---|---|
| Chewi et al. (2021) | $\tilde{\mathcal{O}}\left(\frac{C_{\mathrm{PI}}^{1+1/\alpha} L_\alpha^{2/\alpha} d^{2+1/\alpha}}{\varepsilon^{1/\alpha}}\right)$ | weakly smooth PI, $\alpha > 0$ | Rényi |
| this paper (Thm. 3.5) | $\tilde{\mathcal{O}}\left(C_{\mathrm{PI}} L_\alpha^{2/(1+\alpha)} d^2\right)$ | semi-smooth PI | Rényi |
| Nguyen et al. (2021) | $\tilde{\mathcal{O}}\left(C_{\mathrm{LSI}}^{1+\max\{1/\alpha_i\}} \left[\frac{n\max\{L_{\alpha_i}^2\}d}{\varepsilon}\right]^{\max\{1/\alpha_i\}}\right)$ | $\alpha_i > 0$, composite semi-smooth, LSI | KL |
| this paper (Thm. 4.4) | $\tilde{\mathcal{O}}\left(C_{\mathrm{LSI}} \sum_{i=1}^n L_{\alpha_i}^{2/(\alpha_i+1)} d\right)$ | composite semi-smooth, LSI | KL |
| this paper (Thm. 4.5) | $\tilde{\mathcal{O}}\left(C_{\mathrm{PI}} \sum_{i=1}^n L_{\alpha_i}^{2/(\alpha_i+1)} d\right)$ | composite semi-smooth, PI | Rényi |

**Related works:** MCMC sampling from non-convex potentials has been investigated in Raginsky et al. (2017); Vempala & Wibisono (2019); Wibisono (2019); Chewi et al. (2021); Erdogdu & Hosseinzadeh (2021); Mou et al. (2022b); Luu et al. (2021). There have also been some works on sampling without smoothness Lehec (2021); Durmus et al. (2019); Chatterji et al. (2020); Chewi et al. (2021); Mou et al. (2022a); Shen et al. (2020); Durmus et al. (2018); Freund et al. (2022); Salim & Richtárik (2020); Bernton (2018); Nguyen et al. (2021); Liang & Chen (2022). The literature for the case where the potential function lacks both convexity and smoothness is rather scarce. In Nguyen et al. (2021); Erdogdu & Hosseinzadeh (2021), the authors analyze the convergence of LMC for weakly smooth potentials that satisfy a dissipativity condition. The target distribution is assumed to satisfy some functional inequality. Erdogdu & Hosseinzadeh (2021) also assumes an additional tail growth condition. The dissipativity condition is removed in Chewi et al.

(2021). The results in Nguyen et al. (2021) are applicable to potentials that are the summation of multiple weakly smooth functions, while those in Chewi et al. (2021); Erdogdu & Hosseinzadeh (2021) are not. To compare our results with them, we make the simplification that the initial distance, either in KL or Rényi divergence, to the target distribution is $\tilde{\mathcal{O}}(d)$. The results in cases with non-convex potentials are presented in Table 1. It can be seen that our complexities have better dependence on all the parameters: LSI constant $C_{\mathrm{LSI}}$, PI constant $C_{\mathrm{PI}}$, weakly smooth coefficients $L_\alpha$, and dimension $d$. Moreover, our complexity bounds depend polylogarithmically on the accuracy $\varepsilon$ (thus $\varepsilon$ does not appear in the $\tilde{\mathcal{O}}$ notation) while all the other results have polynomial dependence on $1/\varepsilon$.

## 2 Problem formulation and Background

We are interested in sampling from distributions with potentials that are not necessarily smooth. More specifically, we consider the sampling task with the target distribution $\nu \propto \exp(-f(x))$ where the potential $f$ satisfies

$$\|f'(u) - f'(v)\| \le \sum_{i=1}^{n} L_{\alpha_i}\|u - v\|^{\alpha_i}, \quad \forall u, v \in \mathbb{R}^d \tag{1}$$

for some $\alpha_i \in [0, 1]$ and $L_{\alpha_i} > 0$, $1 \le i \le n$. Here $f'$ denotes a subgradient of $f$ in the Frechet subdifferential (see Definition A.1). When $n = 1$ and $\alpha_1 = 1$, it is well-known that $f$ is a smooth function. When $n = 1$ and $0 < \alpha_1 < 1$, $f$ satisfying (1) is known as a weakly smooth function. When $n = 1$ and $\alpha_1 = 0$, $f$ satisfying (1) is a non-smooth function. Thus, the cases we consider cover all three cases: smooth, weakly-smooth, and non-smooth. For ease of reference, we termed a function satisfying the condition (1) a semi-smooth function. The target distribution $\nu$ is assumed to satisfy LSI or PI, but $f$ can be non-convex. The class of distributions we consider have been studied in (Chatterji et al., 2020; Chewi et al., 2021; Nguyen et al., 2021) for MCMC sampling. One example of such a distribution is $\nu \propto \exp(-\|A_2\sigma(A_1 x) - b\|^2 - \|x\|^2 - \|x\|)$ for some full rank matrices $A_1, A_2$, vector $b$, and some activation (e.g., ReLU) function $\sigma$. This is a Bayesian regression problem.

In this work, we aim to develop a proximal algorithm for sampling from non-convex potentials that satisfy (1). Our method is built on the alternating sampling framework (ASF) introduced in Lee et al. (2021). Unlike most existing sampling algorithms that require the potential to be smooth, ASF is applicable to semi-smooth problems. Initialized at $x_0 \sim \rho_0^X$, ASF with target distribution $\pi^X(x) \propto \exp(-f(x))$ and stepsize $\eta > 0$ performs the two alternating steps as follows.

---

**Algorithm 1** Alternating Sampling Framework (Lee et al., 2021)

---

1. Sample $y_k \sim \pi^{Y|X}(y \mid x_k) \propto \exp[-\frac{1}{2\eta}\|x_k - y\|^2]$
2. Sample $x_{k+1} \sim \pi^{X|Y}(x \mid y_k) \propto \exp[-f(x) - \frac{1}{2\eta}\|x - y_k\|^2]$

---

The ASF is a special instance of the Gibbs sampling (Geman & Geman, 1984) from

$$\pi(x, y) \propto \exp\left(-f(x) - \frac{1}{2\eta}\|x - y\|^2\right). \tag{2}$$

In Algorithm 1, sampling $y_k$ given $x_k$ in step 1 is easy since $\pi^{Y|X}(y \mid x_k) = \mathcal{N}(x_k, \eta I)$ is an isotropic Gaussian distribution. Sampling $x_{k+1}$ given $y_k$ in step 2 is however a highly nontrivial task; it corresponds to the restricted Gaussian oracle for $f$ (Lee et al., 2021), defined as follows.

**Definition 2.1** *Given a point $y \in \mathbb{R}^d$ and stepsize $\eta > 0$, the restricted Gaussian oracle (RGO) for $f : \mathbb{R}^d \to \mathbb{R}$ is a sampling oracle that returns a random sample from a distribution proportional to $\exp(-f(\cdot) - \|\cdot - y\|^2/(2\eta))$.*

The RGO can be viewed as the sampling counterpart of the proximal map in optimization that is widely used in proximal algorithms for optimization (Parikh & Boyd, 2014). The ASF is an idealized algorithm;

an efficient implementation of the RGO is crucial to use this framework in practice. For some special cases of $f$, the RGO admits a computationally efficient realization (Mou et al., 2022a; Shen et al., 2020; Liang & Chen, 2022). For general $f$, especially semi-smooth ones considered in this work, it was not clear how to realize the RGO efficiently.

Under the assumption that the RGO in the ASF can be efficiently realized, the ASF exhibits surprising convergence properties. It was firstly established in Lee et al. (2021) that, when $f$ is strongly convex, Algorithm 1 converges linearly. This result is further improved recently in Chen et al. (2022) under various weaker assumptions on the target distribution $\pi^X \propto \exp(-f)$. We summarize below several convergence results established in Chen et al. (2022) that will be used.

Throughout the paper, we abuse notation by identifying a probability measure with its density w.r.t. Lebesgue measure. To this end, for two probability distributions $\rho \ll \nu$, we denote by

$$H_\nu(\rho) := \int \rho \log \frac{\rho}{\nu}, \quad \chi^2_\nu(\rho) := \int \frac{\rho^2}{\nu} - 1, \quad R_{q,\nu}(\rho) := \frac{1}{q-1} \log \int \frac{\rho^q}{\nu^{q-1}}$$

the *KL divergence*, the *Chi-squared divergence*, and the *Rényi divergence*, respectively. Note that $R_{2,\nu} = \log(1 + \chi^2_\nu)$, $R_{1,\nu} = H_\nu$, and $R_{q,\nu} \leq R_{q',\nu}$ for any $1 \leq q \leq q' < \infty$ (Rényi, 1961; Vempala & Wibisono, 2019). We denote by $W_2$ the Wasserstein-2 distance (Villani, 2021). Recall a probability distribution $\nu$ satisfies LSI with constant $C_{\text{LSI}} > 0$ ($1/C_{\text{LSI}}$-LSI) if for every $\rho$, $H_\nu(\rho) \leq \frac{C_{\text{LSI}}}{2} J_\nu(\rho)$, where the Fisher information $J_\nu(\rho)$ is defined as $J_\nu(\rho) = \mathbb{E}_\rho[\|\nabla \log \frac{\rho}{\nu}\|^2]$. A probability distribution $\nu$ satisfies PI with constant $C_{\text{PI}} > 0$ ($1/C_{\text{PI}}$-PI) if for any smooth bounded function $\psi : \mathbb{R}^d \to \mathbb{R}$, we have $\mathbb{E}_\nu[(\psi - \mathbb{E}_\nu(\psi))^2] \leq C_{\text{PI}}\mathbb{E}_\nu[\|\nabla \psi\|^2]$.

**Theorem 2.2 ((Chen et al., 2022, Theorem 3))** *Assume that $\pi^X \propto \exp(-f)$ satisfies $\lambda$-LSI. For any initial distribution $\rho_0^X$, the $k$-th iterate $\rho_k^X$ of Algorithm 1 with step size $\eta > 0$ satisfies*

$$H_{\pi^X}(\rho_k^X) \leq \frac{H_{\pi^X}(\rho_0^X)}{(1+\lambda\eta)^{2k}}.$$

*Moreover, for all $q \geq 1$,*

$$R_{q,\pi^X}(\rho_k^X) \leq \frac{R_{q,\pi^X}(\rho_0^X)}{(1+\lambda\eta)^{2k/q}}.$$

**Theorem 2.3 ((Chen et al., 2022, Theorem 4))** *Assume $\pi^X \propto \exp(-f)$ satisfies $\lambda$-PI. For any initial distribution $\rho_0^X$, the $k$-th iterate $\rho_k^X$ of Algorithm 1 with step size $\eta > 0$ satisfies*

$$\chi^2_{\pi^X}(\rho_k^X) \leq \frac{\chi^2_{\pi^X}(\rho_0^X)}{(1+\lambda\eta)^{2k}}.$$

*Moreover, for all $q \geq 2$,*

$$R_{q,\pi^X}(\rho_k^X) \leq \begin{cases} R_{q,\pi^X}(\rho_0^X) - \frac{2k\log(1+\lambda\eta)}{q}, & \text{if } k \leq \frac{q}{2\log(1+\lambda\eta)}\left(R_{q,\pi^X}(\rho_0^X) - 1\right), \\ 1/(1+\lambda\eta)^{2(k-k_0)/q}, & \text{if } k \geq k_0 := \left\lceil \frac{q}{2\log(1+\lambda\eta)}\left(R_{q,\pi^X}(\rho_0^X) - 1\right)\right\rceil. \end{cases}$$

**Theorem 2.4 ((Chen et al., 2022, Theorem 2))** *Assume that $\pi^X \propto \exp(-f)$ is log-concave. For any initial distribution $\rho_0^X$, the $k$-th iterate $\rho_k^X$ of Algorithm 1 with step size $\eta > 0$ satisfies*

$$H_{\pi^X}(\rho_k^X) \leq \frac{W_2^2(\rho_0^X, \pi^X)}{k\eta}.$$

To use the ASF for sampling problems, we need to realize the RGO with efficient implementations. In the rest of this paper, we develop an efficient algorithm for RGO associated with a potential satisfying (1), and then combine it with the ASF to establish a proximal algorithm for sampling. The complexity of the proximal algorithm can be obtained by combining the above convergence results for ASF and the complexity results we establish for RGO. The rest of the paper is organized as follows. In Section 3 we consider a special

case of (1) with $n = 1$, develop an efficient implementation for RGO via rejection sampling, and establish complexity results for sampling from distributions with non-convex and semi-smooth potentials. In Section 4, we extend the aforementioned complexity results to the general cases (1). In Section 5, we specialize (1) to the case where $f$ is convex. In Section 6, we present preliminary computational results to demonstrate the efficacy of the proximal sampling algorithm. In Section 7, we present some concluding remarks. Finally, in Appendices A-E, we present technical results and proofs omitted in the paper and provide a self-contained discussion on solving a subproblem in the proximal sampling algorithm.

## 3 Proximal sampling for non-convex and semi-smooth potentials

Our main objective in this section is to establish complexity results for sampling from distributions with non-convex and semi-smooth potentials satisfying (1) with $n = 1$, i.e.,

$$\|f'(u) - f'(v)\| \leq L_\alpha \|u - v\|^\alpha, \quad \forall u, v \in \mathbb{R}^d. \tag{3}$$

We refer to such a function an $L_\alpha$-$\alpha$-semi-smooth function.

The bottleneck of Algorithm 1 for sampling from a general distribution $\exp(-f)$ is an efficient realization of the RGO, i.e., step 2 of Algorithm 1. To address this issue, we develop an efficient algorithm for the corresponding RGO based on rejection sampling. We show that, with a carefully designed proposal and a sufficiently small $\eta$, the expected number of rejection sampling steps to obtain one effective sample in RGO turns out to be bounded above by a dimension-free constant. The core to achieving such a constant bound on the expected rejection steps is a novel construction of proposal distribution that does not rely on the convexity of $f$ and a refined analysis that captures the nature of semi-smooth functions. We utilize a useful property of semi-smooth functions that they can be approximated by smooth functions to arbitrary accuracy, at the cost of increasing their smoothness parameters. This is formalized in the following lemma, of which the proof is postponed to Appendix B. Relevant ideas have been explored in Devolder et al. (2014); Nesterov (2015) to design universal methods for convex semi-smooth optimization problems.

**Lemma 3.1** *Assume $f$ is an $L_\alpha$-$\alpha$-semi-smooth function, then for $\delta > 0$ and every $u, v \in \mathbb{R}^d$*

$$|f(u) - f(v) - \langle f'(v), u - v \rangle| \leq \frac{M}{2} \|u - v\|^2 + \frac{(1-\alpha)\delta}{2}, \tag{4}$$

*where*

$$M = \frac{L_\alpha^{\frac{2}{\alpha+1}}}{[(\alpha+1)\delta]^{\frac{1-\alpha}{\alpha+1}}}. \tag{5}$$

Our algorithm is inspired by Liang & Chen (2022), which also uses rejection sampling for RGO. The proposal of rejection sampling used in Liang & Chen (2022) is a Gaussian distribution centered at an (approximate) minimizer of the regularized potential function. We thus consider the regularized optimization problem

$$\text{prox}_{\eta f}(y) := \operatorname{argmin}_{x \in \mathbb{R}^d} \left\{ f_y^\eta(x) := f(x) + \frac{1}{2\eta} \|x - y\|^2 \right\}, \tag{6}$$

where $y \in \mathbb{R}^d$ is given. Departing from the convex setting studied in Liang & Chen (2022), when $f$ is non-convex and semi-smooth, (6) may not be a convex optimization regardless of the value of $\eta$. Nevertheless, thanks to Lemma 3.1, $f_y^\eta$ is close to a strongly convex and smooth function up to some approximation error, and (6) can still be solved efficiently using convex smooth optimization algorithms such as Nesterov's acceleration. We describe a variant of the method in Algorithm 3 in Appendix D. The following proposition presents a complexity result of Algorithm 3 for finding an approximate stationary point of $f_y^\eta$ with a small $\eta$. Its proof is postponed to Appendix B.

**Proposition 3.2** *Assume $\eta \leq \frac{1}{Md}$, and let $w \in \mathbb{R}^d$ be an approximate stationary point of $f_y^\eta$, i.e.,*

$$\|s\| \leq \sqrt{Md}, \quad s = f'(w) + \frac{1}{\eta}(w - y), \tag{7}$$

*where $M$ is as in (5). Then, the iteration-complexity to find $w$ with Algorithm 3 is $\tilde{\mathcal{O}}(1)$.*

With this approximate stationary point, we obtain the following key ingredients of our rejection sampling-based RGO.

**Lemma 3.3** *Let $w^* \in \mathbb{R}^d$ be a stationary point of $f_y^\eta$, i.e.,*

$$f'(w^*) + \frac{1}{\eta}(w^* - y) = 0, \tag{8}$$

*and $w$ be an approximate stationary point as in (7). Define*

$$h_{1,y}^w(x) := f(w) + \langle f'(w), x - w \rangle - \frac{M}{2}\|x - w\|^2 + \frac{1}{2\eta}\|x - y\|^2 - \frac{(1-\alpha)\delta}{2}, \tag{9}$$

$$h_{2,y}^{w^*}(x) := f(w^*) + \langle f'(w^*), x - w^* \rangle + \frac{M}{2}\|x - w^*\|^2 + \frac{1}{2\eta}\|x - y\|^2 + \frac{(1-\alpha)\delta}{2}, \tag{10}$$

*Then, we have for every $x \in \mathbb{R}^d$,*

$$h_{1,y}^w(x) \le f_y^\eta(x) \le h_{2,y}^{w^*}(x). \tag{11}$$

**Proof**: Inequalities in (11) directly follow from (4) and the definitions of $h_{1,y}^w$ and $h_{2,y}^{w^*}$ in (9) and (10), respectively. ∎

We are now ready to present the rejection sampling algorithm (Algorithm 2) for RGO.

---

**Algorithm 2** RGO Rejection Sampling

---

1. Compute an approximate solution $w$ satisfying (7) with Algorithm 3
2. Generate sample $X \sim \exp(-h_{1,y}^w(x))$
3. Generate sample $U \sim \mathcal{U}[0,1]$
4. If

$$U \le \frac{\exp(-f_y^\eta(X))}{\exp(-h_{1,y}^w(X))},$$

then accept/return $X$; otherwise, reject $X$ and go to step 2.

---

By construction, the proposal $\propto \exp(-h_{1,y}^w(x))$ is close to the target $\pi^{X|Y}(x \mid y) \propto \exp(-f_y^\eta(x))$ when $\eta$ is sufficiently small, and hence the expected number of rejection steps is small. The following proposition rigorously justifies this intuition.

**Proposition 3.4** *Assume $f$ is $L_\alpha$-$\alpha$-semi-smooth and let $f_y^\eta$ be as in (6). Then $X$ generated by Algorithm 2 follows the distribution $\pi^{X|Y}(x \mid y) \propto \exp\left(-f_y^\eta(x)\right)$. Moreover, if*

$$\eta \le \frac{1}{Md} = \frac{[(\alpha+1)\delta]^{\frac{1-\alpha}{\alpha+1}}}{L_\alpha^{\frac{2}{\alpha+1}} d}, \tag{12}$$

*then the expected number of rejection steps in Algorithm 2 is at most $\exp\left(\frac{3(1-\alpha)\delta}{2} + 3\right)$.*

**Proof**: It is well-known in rejection sampling $X \sim \pi^{X|Y}(x|y)$. By definition, the probability that $X$ is accepted is

$$\mathbb{P}\left(U \le \frac{\exp(-f_y^\eta(X))}{\exp(-h_{1,y}^w(X))}\right) = \int \frac{\exp(-f_y^\eta(x))}{\exp(-h_{1,y}^w(x))} \frac{\exp(-h_{1,y}^w(x))}{\int \exp(-h_{1,y}^w(z))\mathrm{d}z}\mathrm{d}x = \frac{\int \exp(-f_y^\eta(x))\mathrm{d}x}{\int \exp(-h_{1,y}^w(x))\mathrm{d}x}.$$

Using the above identity, (11), and Lemma A.4, we have

$$\mathbb{P}\left(U \le \frac{\exp(-f_y^\eta(X))}{\exp(-h_{1,y}^w(X))}\right) \ge \frac{\int \exp(-h_{2,y}^{w^*}(x))\mathrm{d}x}{\int \exp(-h_{1,y}^w(x))\mathrm{d}x}$$

$$= \left(\frac{1-\eta M}{1+\eta M}\right)^{d/2} \frac{\exp\left(\frac{1}{2\eta}\|w^*\|^2 - f(w^*) + \langle f'(w^*), w^*\rangle - \frac{1}{2\eta}\|y\|^2 - \frac{1-\alpha}{2}\delta\right)}{\exp\left(\frac{1}{2\eta}\|w\|^2 + \frac{\eta}{2(1-\eta M)}\|s\|^2 - f(w) + \frac{1}{\eta}\langle w, y-w\rangle - \frac{1}{2\eta}\|y\|^2 + \frac{1-\alpha}{2}\delta\right)}$$

$$= \left(\frac{1-\eta M}{1+\eta M}\right)^{d/2} \exp\left(-(1-\alpha)\delta - \frac{\eta}{2(1-\eta M)}\|s\|^2\right)$$

$$\times \exp\left(\frac{1}{2\eta}\|w^*\|^2 - \frac{1}{2\eta}\|w\|^2 - f(w^*) + f(w) + \langle f'(w^*), w^*\rangle - \frac{1}{\eta}\langle w, y-w\rangle\right). \tag{13}$$

It follows from (4) with $(u,v) = (w, w^*)$ that

$$-f(w) + f(w^*) + \langle f'(w^*), w - w^*\rangle \le \frac{M}{2}\|w - w^*\|^2 + \frac{(1-\alpha)\delta}{2},$$

which together with (8) implies that

$$\frac{1}{2\eta}\|w^*\|^2 - \frac{1}{2\eta}\|w\|^2 - f(w^*) + f(w) + \langle f'(w^*), w^*\rangle - \frac{1}{\eta}\langle w, y-w\rangle$$

$$= \frac{1}{2\eta}\|w^*\|^2 - \frac{1}{2\eta}\|w\|^2 - f(w^*) + f(w) + \langle f'(w^*), w^*\rangle - \frac{1}{\eta}\langle w, \eta f'(w^*) + w^* - w\rangle$$

$$= \frac{1}{2\eta}\|w - w^*\|^2 - f(w^*) + f(w) + \langle f'(w^*), w^* - w\rangle$$

$$\ge \frac{1}{2\eta}\|w - w^*\|^2 - \frac{M}{2}\|w - w^*\|^2 - \frac{(1-\alpha)\delta}{2} \ge -\frac{(1-\alpha)\delta}{2},$$

where the last inequality is due to (12). Plugging the above inequality into (13), we obtain

$$\mathbb{P}\left(U \le \frac{\exp(-f_y^\eta(X))}{\exp(-h_{1,y}^w(X))}\right) \ge \left(\frac{1-\eta M}{1+\eta M}\right)^{d/2} \exp\left(-\frac{3(1-\alpha)\delta}{2} - \frac{\eta}{2(1-\eta M)}\|s\|^2\right).$$

Hence, using the above bound, (12) and (7), we arrive at the following bound on the expected number of rejection iterations

$$\frac{1}{\mathbb{P}\left(U \le \frac{\exp(-f_y^\eta(X))}{\exp(-h_{1,y}^w(X))}\right)} \le \left(\frac{1+\eta M}{1-\eta M}\right)^{d/2} \exp\left(\frac{3(1-\alpha)\delta}{2} + \frac{\eta}{2(1-\eta M)}\|s\|^2\right)$$

$$\le \left(1 + \frac{2\eta M}{1-\eta M}\right)^{d/2} \exp\left(\frac{3(1-\alpha)\delta}{2} + \eta\|s\|^2\right) \le (1 + 4\eta M)^{d/2} \exp\left(\frac{3(1-\alpha)\delta}{2} + \frac{\|s\|^2}{Md}\right)$$

$$\le \left(1 + \frac{4}{d}\right)^{d/2} \exp\left(\frac{3(1-\alpha)\delta}{2} + 1\right) \le \exp\left(\frac{3(1-\alpha)\delta}{2} + 3\right).$$

∎

Note that $\delta$ is a tunable parameter. Choosing a large $\delta$ makes $M$ small and $\eta$ large in view of (5) and (12), respectively. Such a choice results in a better complexity of ASF but a larger number of expected rejection steps in RGO. Combining Proposition (3.4) and the convergence results of ASF we obtain the following non-asymptotic complexity bound to sample from non-convex semi-smooth potentials. This complexity bound is better than all existing results when $\alpha \in [0, 1)$, see Table 1 for comparison.

**Theorem 3.5** *Assume $f$ is $L_\alpha$-$\alpha$-semi-smooth and $\pi^X \propto \exp(-f)$ satisfies PI with constant $C_{\mathrm{PI}}$. With initial distribution $\rho_0^X$ and stepsize $\eta \asymp 1/(L_\alpha^{\frac{2}{\alpha+1}} d)$, Algorithm 1 using Algorithm 2 as an RGO has the*

*iteration-complexity bound*

$$\tilde{\mathcal{O}}\left(C_{\text{PI}}L_\alpha^{\frac{2}{\alpha+1}}d\log\chi_{\pi^X}^2(\rho_0^X)\right) \tag{14}$$

*to achieve $\varepsilon$ error to the target $\pi^X$ in terms of Chi-squared divergence, and*

$$\tilde{\mathcal{O}}\left(C_{\text{PI}}L_\alpha^{\frac{2}{\alpha+1}}qdR_{q,\pi^X}(\rho_0^X)\right), \quad q \geq 2 \tag{15}$$

*to achieve $\varepsilon$ error in Rényi divergence $R_{q,\pi^X}$. Each iteration queries $\tilde{\mathcal{O}}(1)$ subgradients of $f$.*

**Proof**: The result is a direct consequence of Theorem 2.3, Proposition 3.2 and Proposition 3.4 with the choice of stepsize $\eta \asymp 1/(L_\alpha^{\frac{2}{\alpha+1}}d)$. ∎

*Remark*: When the coordinate is scaled by $\gamma$ as $x \to \gamma x$, by definition, the semi-smoothness coefficient becomes $\gamma^{\alpha+1}L_\alpha$ and the PI constant becomes $C_{\text{PI}}/\gamma^2$. By Proposition 3.4, to attain $\mathcal{O}(1)$ complexity for the RGO, the stepsize should be $\eta \leq \frac{[(\alpha+1)\delta]^{\frac{1-\alpha}{\alpha+1}}}{(\gamma^{\alpha+1}L_\alpha)^{\frac{2}{\alpha+1}}d} = \frac{[(\alpha+1)\delta]^{\frac{1-\alpha}{\alpha+1}}}{\gamma^2 L_\alpha^{\frac{2}{\alpha+1}}d}$. Thus, applying Theorem 2.3, the total complexity remains the same as that without coordinate scaling. ∎

## 4 Proximal sampling for non-convex composite potentials

This section is devoted to the sampling from distributions with non-convex composite potential $f$ satisfying (1). Results presented in this section generalize those in Section 3, which are for the setting with $n = 1$. Although Section 3 is developed for the simple case where $n = 1$ in (1), the proof techniques apply to the general case of (1). Hence, to avoid duplication, we present the following results analogous to those in Section 3 without giving proofs.

The following lemma is a direct generalization of Lemma 3.1, which shows that functions satisfying (1) can be approximated by smooth functions up to some controllable approximation errors.

**Lemma 4.1** *Assume $f$ satisfies (1), then for any $\delta > 0$, we have*

$$|f(u) - f(v) - \langle f'(v), u - v\rangle| \leq \frac{M_n}{2}\|u - v\|^2 + \sum_{i=1}^n \frac{(1-\alpha_i)\delta}{2}, \quad \forall u, v \in \mathbb{R}^d, \tag{16}$$

*where*

$$M_n = \sum_{i=1}^n \frac{L_{\alpha_i}^{\frac{2}{\alpha_i+1}}}{[(\alpha_i+1)\delta]^{\frac{1-\alpha_i}{\alpha_i+1}}}. \tag{17}$$

The next proposition is a counterpart of Proposition 3.2 and gives the complexity of solving optimization problem (6) in the context of (1).

**Proposition 4.2** *Assume $\eta \leq 1/(M_nd)$, and let $w_n \in \mathbb{R}^d$ be an approximate stationary point of $f_y^\eta$ such that $\|f'(w_n) + \frac{1}{\eta}(w_n - y)\| \leq \sqrt{M_nd}$. Then, the iteration-complexity to find $w_n$ with Algorithm 3 is $\tilde{\mathcal{O}}(1)$.*

Again, the core to the proximal algorithm (Algorithm 1) is an efficient implementation of RGO. We use Algorithm 2 with slight modification as a realization of RGO in the context of (1). First, in step 1 of Algorithm 2, we use Algorithm 3 to compute $w_n$ as in Proposition 4.2 instead of $w$. Second, in steps 2 and 4 of Algorithm 2, instead of using $h_{1,y}^w$ as in (9), we define $h_{1,y}^w$ as follows,

$$h_{1,y}^w(x) = f(w) + \langle f'(w), x - w\rangle - \frac{M_n}{2}\|x - w\|^2 + \frac{1}{2\eta}\|x - y\|^2 - \sum_{i=1}^n \frac{(1-\alpha_i)\delta}{2},$$

where $M_n$ is as in (17). The next proposition is a generalization of Proposition 3.4.

**Proposition 4.3** *Assume $f$ satisfies (1) and let $f_y^\eta$ be as in (6). Then $X$ generated by Algorithm 2 with modification follows the distribution $\pi^{X|Y}(x \mid y) \propto \exp\left(-f_y^\eta(x)\right)$. Moreover, if $\eta \le \frac{1}{M_n d}$, then the expected number of rejection steps in Algorithm 2 is at most $\exp\left(\frac{3\sum_{i=1}^n(1-\alpha_i)\delta}{2} + 3\right)$.*

In the rest of this section, we present two results on the complexity of sampling for non-convex composite potentials where the target distribution satisfies either LSI or PI. Our complexity bounds are better than all existing results when $\min \alpha_i \in [0,1)$, see Table 1 for comparison.

**Theorem 4.4** *Assume $f$ satisfies (1) and $\pi^X \propto \exp(-f)$ satisfies LSI with constant $C_{\mathrm{LSI}}$. With initial distribution $\rho_0^X$ and stepsize $\eta \asymp 1/\left(\sum_{i=1}^n L_{\alpha_i}^{\frac{2}{\alpha_i+1}} d\right)$, Algorithm 1 using the modified Algorithm 2 as an RGO has the iteration-complexity bound*

$$\tilde{\mathcal{O}}\left(C_{\mathrm{LSI}}\sum_{i=1}^n L_{\alpha_i}^{\frac{2}{\alpha_i+1}} d\right) \tag{18}$$

*to achieve $\varepsilon$ error to $\pi^X$ in KL divergence. Each iteration queries $\tilde{\mathcal{O}}(1)$ subgradients of $f$ and generates $\mathcal{O}(1)$ samples in expectation from Gaussian distribution.. Moreover, for all $q \ge 1$, the iteration-complexity bound to achieve $\varepsilon$ error to $\pi^X$ in Rényi divergence $R_{q,\pi^X}$ is*

$$\tilde{\mathcal{O}}\left(C_{\mathrm{LSI}}\sum_{i=1}^n L_{\alpha_i}^{\frac{2}{\alpha_i+1}} dq\right).$$

**Proof**: This theorem is a direct consequence of Theorem 2.2 and Propositions 4.2-4.3. ∎

**Theorem 4.5** *Assume $f$ satisfies (1) and $\pi^X \propto \exp(-f)$ satisfies PI with constant $C_{\mathrm{PI}}$. With initial distribution $\rho_0^X$ and stepsize $\eta \asymp 1/\left(\sum_{i=1}^n L_{\alpha_i}^{\frac{2}{\alpha_i+1}} d\right)$, Algorithm 1 using the modified Algorithm 2 as an RGO has the iteration-complexity bound*

$$\tilde{\mathcal{O}}\left(C_{\mathrm{PI}}\sum_{i=1}^n L_{\alpha_i}^{\frac{2}{\alpha_i+1}} d\log \chi_{\pi^X}^2(\rho_0^X)\right) \tag{19}$$

*to achieve $\varepsilon$ error to the target $\pi^X$ in terms of Chi-squared divergence, and*

$$\tilde{\mathcal{O}}\left(C_{\mathrm{PI}}\sum_{i=1}^n L_{\alpha_i}^{\frac{2}{\alpha_i+1}} qdR_{q,\pi^X}(\rho_0^X)\right), \quad q \ge 2 \tag{20}$$

*to achieve $\varepsilon$ error in Rényi divergence $R_{q,\pi^X}$. Each iteration queries $\tilde{\mathcal{O}}(1)$ subgradients of $f$ and generates $\mathcal{O}(1)$ samples in expectation from Gaussian distribution.*

**Proof**: This theorem immediately follows from Theorem 2.3 and Propositions 4.2-4.3. ∎

## 5 Proximal sampling for convex composite potentials

For completeness, in this section we present a complexity result for sampling from distributions with convex potential $f$ satisfying (1). We only consider the weakly convex setting here as strongly log-concave distributions satisfy LSI and are thus covered by Theorem 4.4. The result is a consequence of Theorem 2.4 for log-concave densities combined with our RGO implementation. In particular, Lemma 4.1 and Propositions 4.2-4.3 apply to this setting. Hence, using Theorem 2.4 and Propositions 4.2-4.3, we directly obtain the following complexity result. Note our result is applicable to composite potentials with any number of components while most existing results are only applicable to composite potentials with at most two components. We also present an alternative result via a regularization approach in Appendix C.

**Theorem 5.1** *Assume $f$ satisfies (1) and $\pi^X \propto \exp(-f)$ is log-concave. With initial distribution $\rho_0^X$ and stepsize $\eta \asymp 1/\left(\sum_{i=1}^n L_{\alpha_i}^{\frac{2}{\alpha_i+1}} d\right)$, Algorithm 1 using Algorithm 2 as an RGO has the iteration-complexity bound*

$$\tilde{\mathcal{O}}\left(\frac{W_2^2(\rho_0^X, \pi^X)\sum_{i=1}^n L_{\alpha_i}^{\frac{2}{\alpha_i+1}} d}{\varepsilon}\right)$$

*to achieve $\varepsilon$ error to the target $\pi^X$ in terms of KL divergence. Each iteration queries $\tilde{\mathcal{O}}(1)$ subgradients of $f$ and generates $\mathcal{O}(1)$ samples in expectation from standard Gaussian distribution.*

**Proof**: This theorem is a direct consequence of Theorem 2.4 and Propositions 4.2-4.3. ■

## 6 Computational results

In this section, we present a numerical example to illustrate our result. We consider sampling from a Gaussian-Laplace mixture

$$\nu(x) = 0.5(2\pi)^{-d/2}\sqrt{\det Q}\exp(-(x-\mathbf{1})^\top Q(x-\mathbf{1})/2) + 0.5(2^d)\exp(-\|4x\|_1)$$

where $Q = USU^\top$, $d = 5$, $S = \mathrm{diag}(14, 15, 16, 17, 18)$, and $U$ is an arbitrary orthogonal matrix.

We run $500,000$ iterations (with $100,000$ burn-in iterations) for both the proximal sampling algorithm and LMC with $\eta = 1/(Md)$ where $d = 5$ and $M$ is as in (5) with $(\alpha, L_\alpha) = (1, 27)$ and $\delta = 1$. Histograms and trace plots (of the 3-rd coordinate) of the samples generated by both methods are presented in Figures 1 and 2. The average numbers of optimization iterations and rejection sampling iterations in Algorithm 2 are 1.5 and 1.3, respectively. In addition, we also run $2,500,000$ iterations (with $500,000$ burn-in iterations) for LMC with $\eta = 1/(5Md)$. Figure 3 presents the histogram and trace plot for this LMC. The red curves in histograms are the scaled target density $\nu(x)$. Figures 4 and 5 show the results for MALA in the same settings as those of Figures 2 and 3, respectively. In these experiments, we observe that our algorithm achieves better accuracy under the same computational budget constraints.

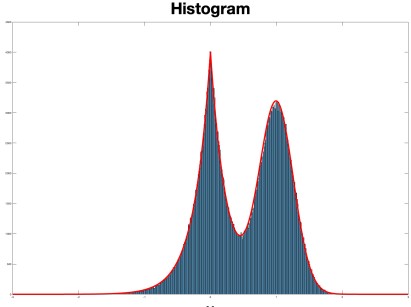 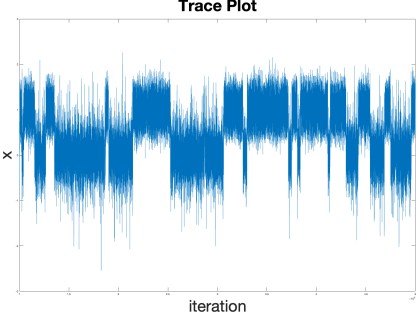

Figure 1: Gaussian-Lasso mixture using the proximal sampling algorithm

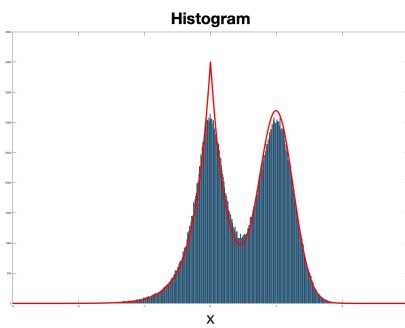
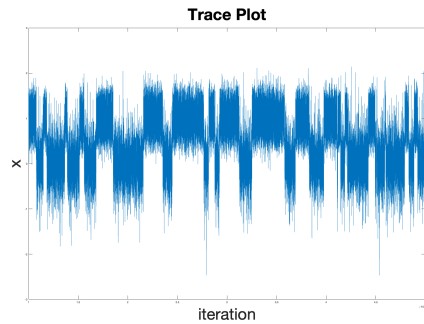

Figure 2: Gaussian-Lasso mixture using LMC

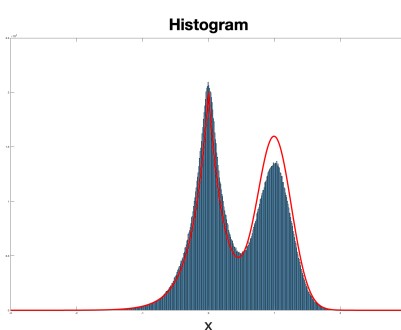
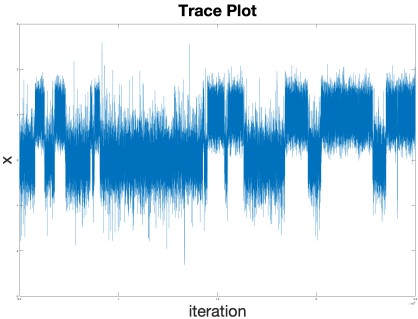

Figure 3: Gaussian-Lasso mixture using LMC with small $\eta$

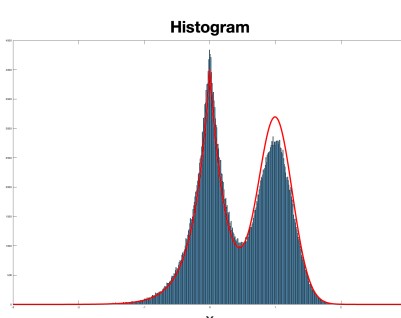
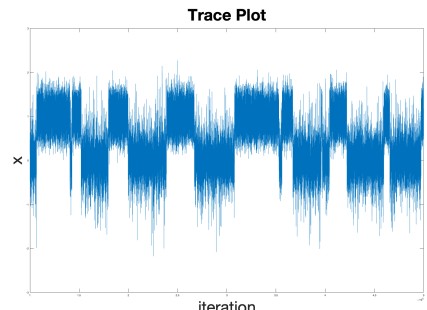

Figure 4: Gaussian-Lasso mixture using MALA

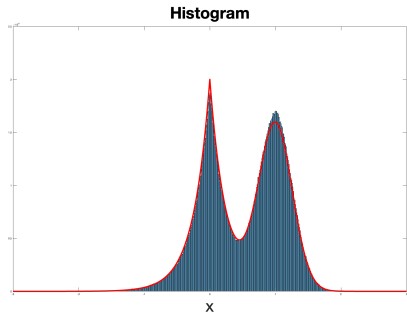
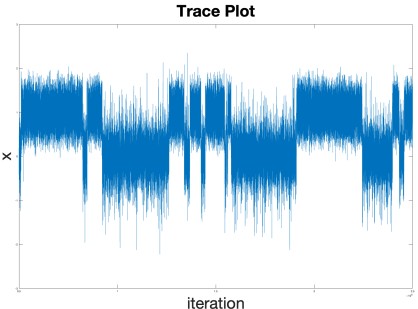

Figure 5: Gaussian-Lasso mixture using MALA with small $\eta$

## 7 Conclusions

In this paper, we develop a novel proximal sampling algorithm for distributions with semi-smooth potentials and establish complexity-bound results for the proposed method under the assumption that distributions are log-concave or satisfy LSI or PI. Our proximal algorithm is based on the ASF, which resembles the proximal point method in optimization. Each iteration of the ASF generates a sample from a regularized target distribution by querying the RGO, which is itself a challenging algorithmic task due to the lack of smoothness and possibly convexity. The core to our approach is an efficient realization of the RGO based on rejection sampling. We develop a novel technique to bound the expected number of rejection steps in the RGO, which leads to competitive complexity results to sample from semi-smooth and possibly non-convex potentials.

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

## Contents

## A   Technical results

This section collects the definition of Frechet subdifferential and a few technical results that are useful for the analysis of RGO in Section 3.

**Definition A.1 (Frechet subdifferential)** *Let $f : \mathbb{R}^n \to \mathbb{R} \cup \{\infty\}$ be a proper closed function, then the Frechet subdifferential is defined as*

$$\tilde{\partial} f(x) = \left\{ v \in \mathbb{R}^n : \liminf_{y \to x} \frac{f(y) - f(x) - \langle v, y - x \rangle}{\|y - x\|} \geq 0 \right\}.$$

The first result is the well-known Gaussian integral.

**Lemma A.2** *A useful Gaussian integral: for any $\eta > 0$,*

$$\int_{\mathbb{R}^d} \exp\left(-\frac{1}{2\eta}\|x\|^2\right) \mathrm{d}x = (2\pi\eta)^{d/2}.$$

The following lemma shows that $f_y^\eta$ as in (6) is close to a strongly convex and smooth function when $\eta$ is small. This result is used in Proposition 3.2.

**Lemma A.3** *Let $f_y^\eta := f + \frac{1}{2\eta}\|\cdot - y\|^2$ and $(f_y^\eta)' := f' + \frac{1}{\eta}(\cdot - y)$, then we have for every $u, v \in \mathbb{R}^d$,*

$$\frac{1}{2}\left(\frac{1}{\eta} - M\right)\|u - v\|^2 - \frac{(1-\alpha)\delta}{2} \leq f_y^\eta(u) - f_y^\eta(v) - \langle (f_y^\eta)'(v), u - v \rangle$$

$$\leq \frac{1}{2}\left(\frac{1}{\eta} + M\right)\|u - v\|^2 + \frac{(1-\alpha)\delta}{2}.$$

**Proof**: Using the definitions of $f_y^\eta$ and $(f_y^\eta)'$, we have

$$f_y^\eta(u) - f_y^\eta(v) - \langle (f_y^\eta)'(v), u - v \rangle$$
$$= f(u) - f(v) - \langle f'(v), u - v \rangle + \frac{1}{2\eta}\|u - y\|^2 - \frac{1}{2\eta}\|v - y\|^2 - \frac{1}{\eta}\langle v - y, u - v \rangle$$
$$= f(u) - f(v) - \langle f'(v), u - v \rangle + \frac{1}{2\eta}\|u - v\|^2.$$

The lemma now follows from above identity and (4). ∎

The next lemma gives equivalent formulas of $\int \exp(-h_{1,y}^w(x))\mathrm{d}x$ and $\int \exp(-h_{2,y}^{w^*}(x))\mathrm{d}x$ that are useful in Proposition 3.4.

**Lemma A.4** *Recall $w$ and $w^*$ are defined in Proposition 3.2 and Lemma 3.3, respectively. Let $h_{1,y}^w$ and $h_{2,y}^{w^*}$ be as in (9) and (10), respectively. Then, we have the following integrals*

$$\int \exp(-h_{1,y}^w(x))\mathrm{d}x = \left(\frac{2\pi\eta}{1 - \eta M}\right)^{d/2} \exp\left(H_1(w)\right), \tag{21}$$

$$\int \exp(-h_{2,y}^{w^*}(x))\mathrm{d}x = \left(\frac{2\pi\eta}{1 + \eta M}\right)^{d/2} \exp\left(H_2(w^*)\right), \tag{22}$$

*where*

$$H_1(w) = \frac{1}{2\eta}\|w\|^2 + \frac{\eta}{2(1 - \eta M)}\|s\|^2 - f(w) + \frac{1}{\eta}\langle w, y - w \rangle - \frac{1}{2\eta}\|y\|^2 + \frac{1-\alpha}{2}\delta, \tag{23}$$

$$H_2(w^*) = \frac{1}{2\eta}\|w^*\|^2 - f(w^*) + \langle f'(w^*), w^* \rangle - \frac{1}{2\eta}\|y\|^2 - \frac{1-\alpha}{2}\delta. \tag{24}$$

**Proof**: We first rewrite $h_{1,y}^w$ and $h_{2,y}^{w^*}$ as follows

$$
\begin{aligned}
h_{1,y}^w(x) =& f(w) + \langle f'(w), x - w \rangle - \frac{M}{2} \|x - w\|^2 + \frac{1}{2\eta} \|x - y\|^2 - \frac{(1-\alpha)\delta}{2} \\
=& \frac{1 - \eta M}{2\eta} \left\| x + \frac{\eta M}{1 - \eta M} w - \frac{1}{1 - \eta M} y + \frac{\eta}{1 - \eta M} f'(w) \right\|^2 \\
& - \frac{1}{2\eta(1 - \eta M)} \|\eta M w - y + \eta f'(w)\|^2 \\
& + f(w) - \langle f'(w), w \rangle - \frac{M}{2} \|w\|^2 + \frac{1}{2\eta} \|y\|^2 - \frac{1 - \alpha}{2} \delta,
\end{aligned}
\tag{25}
$$

and

$$
\begin{aligned}
h_{2,y}^{w^*}(x) =& f(w^*) + \langle f'(w^*), x - w^* \rangle + \frac{M}{2} \|x - w^*\|^2 + \frac{1}{2\eta} \|x - y\|^2 + \frac{(1-\alpha)\delta}{2} \\
=& \frac{1 + \eta M}{2\eta} \left\| x - \frac{\eta M}{1 + \eta M} w^* - \frac{1}{1 + \eta M} y + \frac{\eta}{1 + \eta M} f'(w^*) \right\|^2 \\
& - \frac{1}{2\eta(1 + \eta M)} \|\eta M w^* + y - \eta f'(w^*)\|^2 \\
& + f(w^*) - \langle f'(w^*), w^* \rangle + \frac{M}{2} \|w^*\|^2 + \frac{1}{2\eta} \|y\|^2 + \frac{1 - \alpha}{2} \delta.
\end{aligned}
\tag{26}
$$

It follows from (25) and Lemma A.2 that

$$
\int \exp(-h_{1,y}^w(x)) \mathrm{d}x = \left( \frac{2\pi\eta}{1 - \eta M} \right)^{d/2} \exp(\hat{H}_1(w)),
\tag{27}
$$

$$
\int \exp(-h_{2,y}^{w^*}(x)) \mathrm{d}x = \left( \frac{2\pi\eta}{1 + \eta M} \right)^{d/2} \exp(\hat{H}_2(w^*)),
\tag{28}
$$

where

$$
\begin{aligned}
\hat{H}_1(w) =& \frac{1}{2\eta(1 - \eta M)} \|\eta M w - y + \eta f'(w)\|^2 - f(w) + \langle f'(w), w \rangle \\
& + \frac{M}{2} \|w\|^2 - \frac{1}{2\eta} \|y\|^2 + \frac{1 - \alpha}{2} \delta, \\
\hat{H}_2(w^*) =& \frac{1}{2\eta(1 + \eta M)} \|\eta M w^* + y - \eta f'(w^*)\|^2 - f(w^*) + \langle f'(w^*), w^* \rangle \\
& - \frac{M}{2} \|w^*\|^2 - \frac{1}{2\eta} \|y\|^2 - \frac{1 - \alpha}{2} \delta.
\end{aligned}
$$

It suffices to show that $\hat{H}_1(w) = H_1(w)$ and $\hat{H}_2(w^*) = H_2(w^*)$ to complete the proof.

We first verify that $\hat{H}_1(w) = H_1(w)$. Using the definition of $\hat{H}_1(w)$ above and the definition of $s$ in (7), we have

$$
\begin{aligned}
\hat{H}_1(w) =& \frac{1}{2\eta(1-\eta M)}\|\eta Mw - w + w - y + \eta f'(w)\|^2 - f(w) + \langle f'(w), w\rangle \\
&+ \frac{M}{2}\|w\|^2 - \frac{1}{2\eta}\|y\|^2 + \frac{1-\alpha}{2}\delta \\
=& \frac{1}{2\eta(1-\eta M)}\|(\eta M - 1)w + \eta s\|^2 - f(w) + \langle f'(w), w\rangle \\
&+ \frac{M}{2}\|w\|^2 - \frac{1}{2\eta}\|y\|^2 + \frac{1-\alpha}{2}\delta \\
=& \frac{1-\eta M}{2\eta}\|w\|^2 + \frac{M}{2}\|w\|^2 - \langle w, s\rangle + \frac{\eta}{2(1-\eta M)}\|s\|^2 - f(w) + \langle f'(w), w\rangle \\
&- \frac{1}{2\eta}\|y\|^2 + \frac{1-\alpha}{2}\delta \\
=& \frac{1}{2\eta}\|w\|^2 + \frac{\eta}{2(1-\eta M)}\|s\|^2 - f(w) + \frac{1}{\eta}\langle w, y - w\rangle - \frac{1}{2\eta}\|y\|^2 + \frac{1-\alpha}{2}\delta.
\end{aligned}
$$

In view of (23), we verify that $\hat{H}_1(w) = H_1(w)$ and hence (21) is proved.

We next verify that $\hat{H}_2(w^*) = H_2(w^*)$. Using the definition of $\hat{H}_2(w^*)$ and (8), we have

$$
\begin{aligned}
\hat{H}_2(w^*) =& \frac{1}{2\eta(1+\eta M)}\|\eta Mw^* + w^* - w^* + y - \eta f'(w^*)\|^2 - f(w^*) + \langle f'(w^*), w^*\rangle \\
&- \frac{M}{2}\|w^*\|^2 - \frac{1}{2\eta}\|y\|^2 - \frac{1-\alpha}{2}\delta \\
=& \frac{1}{2\eta(1+\eta M)}\|(\eta M + 1)w^*\|^2 - \frac{M}{2}\|w^*\|^2 - f(w^*) + \langle f'(w^*), w^*\rangle \\
&- \frac{1}{2\eta}\|y\|^2 - \frac{1-\alpha}{2}\delta \\
=& \frac{1}{2\eta}\|w^*\|^2 - f(w^*) + \langle f'(w^*), w^*\rangle - \frac{1}{2\eta}\|y\|^2 - \frac{1-\alpha}{2}\delta.
\end{aligned}
$$

In view of (24), we verify that $\hat{H}_2(w^*) = H_2(w^*)$ and hence (22) is proved. ∎

## B  Missing proofs

### B.1  Proof of Lemma 3.1

It follows from the assumption that $f$ is $L_\alpha$-semi-smooth that for every $u, v \in \mathbb{R}^d$,

$$
|f(u) - f(v) - \langle f'(v), u - v\rangle| \le \frac{L_\alpha}{\alpha + 1}\|u - v\|^{\alpha+1}. \tag{29}
$$

Using the Young's inequality $ab \le a^p/p + b^q/q$ with

$$
a = \frac{L_\alpha}{(\alpha+1)\delta^{\frac{1-\alpha}{2}}}\|u - v\|^{\alpha+1}, \quad b = \delta^{\frac{1-\alpha}{2}}, \quad p = \frac{2}{\alpha+1}, \quad q = \frac{2}{1-\alpha},
$$

we obtain

$$
\frac{L_\alpha}{\alpha + 1}\|u - v\|^{\alpha+1} \le \frac{L_\alpha^{\frac{2}{\alpha+1}}}{2[(\alpha+1)\delta]^{\frac{1-\alpha}{\alpha+1}}}\|u - v\|^2 + \frac{(1-\alpha)\delta}{2}.
$$

Plugging the above inequality into (29), we have

$$
|f(u) - f(v) - \langle f'(v), u - v\rangle| \le \frac{L_\alpha^{\frac{2}{\alpha+1}}}{2[(\alpha+1)\delta]^{\frac{1-\alpha}{\alpha+1}}}\|u - v\|^2 + \frac{(1-\alpha)\delta}{2}.
$$

This inequality and the definition of $M$ in (5) imply (4). ∎

## B.2   Proof of Proposition 3.2

It follows from Lemma A.3 that $f_y^\eta$ satisfies (38) with

$$\mu = \frac{1}{\eta} - M, \quad L = \frac{1}{\eta} + M, \quad \theta = \frac{(1-\alpha)\delta}{2}. \tag{30}$$

Since $\eta \leq 1/(Md)$, it is easy to verify that the assumption on $\rho$ (i.e., $\sqrt{Md}$ in our case) in Proposition D.4 is satisfied. Hence, it follows from Proposition D.4 and (30) that the proposition holds. ∎

## C   Sampling from regularized convex semi-smooth potentials

This section presents an alternative approach via regularization for sampling in the same setting as in Section 5, i.e., $f$ is convex and satisfies (1). More specifically, the alternative complexity result is obtained by first applying Theorem 2.2 on a regularized convex semi-smooth potential and then specifying the regularization parameter.

We consider the following regularized potential with some $\mu > 0$,

$$\hat{f}(\cdot) = f(\cdot) + \frac{\mu}{2}\|\cdot - x^0\|^2, \tag{31}$$

which is clearly $\mu$-strongly convex by construction. Hence, $\exp(-\hat{f})$ satisfies $\mu$-LSI and Theorem 2.2 is applicable. Since $\hat{f}$ is $\mu$-strongly convex, improved versions of results in Section 4 can be obtained. We omit the proofs since they can be easily obtained by following similar ideas in the proofs given in Section 3.

**Lemma C.1** *Assume $f$ is a convex function and satisfies (1), then for $\delta > 0$ and every $u, v \in \mathbb{R}^d$, we have*

$$\hat{f}(u) - \hat{f}(v) - \langle \hat{f}'(v), u - v \rangle \leq \frac{M_n + \mu}{2}\|u - v\|^2 + \sum_{i=1}^n \frac{(1-\alpha_i)\delta}{2},$$

$$\hat{f}(u) - \hat{f}(v) - \langle \hat{f}'(v), u - v \rangle \geq \frac{\mu}{2}\|u - v\|^2,$$

*where $\hat{f}$ and $M_n$ are as in (31) and (17), respectively.*

**Proposition C.2** *Assume $\eta \leq \frac{1}{M_n d}$, and let $w \in \mathbb{R}^d$ be an approximate stationary point of*

$$\min_{x \in \mathbb{R}^d} \left\{ \hat{f}_y^\eta(x) := \hat{f}(x) + \frac{1}{2\eta}\|x - y\|^2 \right\}, \tag{32}$$

*i.e.,*

$$\|s\| \leq \sqrt{M_n d}, \quad s = \hat{f}'(w) + \frac{1}{\eta}(w - y), \tag{33}$$

*where $M_n$ is as in (17). Then, the iteration-complexity to find $w$ by using Algorithm 3 is $\tilde{\mathcal{O}}(1)$.*

**Lemma C.3** *Let $w^* \in \mathbb{R}^d$ be a stationary point of $\hat{f}_y^\eta$, i.e.,*

$$\hat{f}'(w^*) + \frac{1}{\eta}(w^* - y) = 0.$$

*Define*

$$\hat{h}_{1,y}^w(x) := \hat{f}(w) + \langle \hat{f}'(w), x - w \rangle + \frac{\mu}{2}\|x - w\|^2 + \frac{1}{2\eta}\|x - y\|^2,$$

$$\hat{h}_{2,y}^{w^*}(x) := \hat{f}(w^*) + \langle \hat{f}'(w^*), x - w^* \rangle + \frac{M_n + \mu}{2}\|x - w^*\|^2 + \frac{1}{2\eta}\|x - y\|^2 + \sum_{i=1}^n \frac{(1-\alpha_i)\delta}{2},$$

*where $w$ is as in (33). Then, we have for every $x \in \mathbb{R}^d$,*

$$\hat{h}_{1,y}^w(x) \le \hat{f}_y^\eta(x) \le \hat{h}_{2,y}^{w^*}(x).$$

**Proposition C.4** *Assume $f$ is convex and satisfies (1) and let $\hat{f}_y^\eta$ be as in (32). Then $X$ generated by Algorithm 2 (with (7), $f_y^\eta$, and $h_{1,y}^w(x)$ replaced by (33), $\hat{f}_y^\eta$, and $\hat{h}_{1,y}^w(x)$, respectively) follows the distribution $\pi^{X|Y}(x \mid y) \propto \exp\left(-\hat{f}_y^\eta(x)\right)$. Moreover, if $\eta \le \frac{1}{M_n d}$, then the expected number of rejection steps in Algorithm 2 is at most $\exp\left(\sum_{i=1}^n \frac{(1-\alpha_i)\delta}{2} + 1\right)$.*

**Proposition C.5** *Assume $f$ is convex and satisfies (1), and let $\hat{f}$ be as in (31). With initial distribution $\rho_0^X$ and stepsize $\eta \asymp 1/\left(\sum_{i=1}^n L_{\alpha_i}^{\frac{2}{\alpha_i+1}} d\right)$, Algorithm 1 using Algorithm 2 as an RGO achieves $\varepsilon$ error in terms of KL divergence with respect to $\exp(-\hat{f})$ in*

$$\tilde{\mathcal{O}}\left(\frac{\sum_{i=1}^n L_{\alpha_i}^{\frac{2}{\alpha_i+1}} d}{\mu}\right) \tag{34}$$

*iteration, and each iteration queries $\tilde{\mathcal{O}}(1)$ subgradient oracle of $f$ and $\mathcal{O}(1)$ Gaussian distribution sampling oracle.*

**Proof**: By construction, $\hat{f}$ is $\mu$-strongly convex and $\exp(-\hat{f})$ satisfies $\mu$-LSI. Using Theorem 2.2, starting from initial distribution $\rho_0^X$, it takes

$$\frac{1}{2\mu\eta} \log \frac{H_{\pi^X}(\rho_0^X)}{\varepsilon} \tag{35}$$

iterations for Algorithm 1 to achieve $\varepsilon$ error to $\exp(-\hat{f})$ with respect to KL divergence.

By Propositions C.2 and C.4, Algorithm 2 queries $\tilde{\mathcal{O}}(1)$ subgradient oracle of $f$ and $\mathcal{O}(1)$ Gaussian distribution sampling oracle. Complexity (34) then follows by plugging $\eta$ into (35). ∎

Building upon Proposition C.5 for sampling from $\exp(-\hat{f}(x)) = \exp(-f(x) - \mu\|x - x^0\|^2/2)$, we establish a complexity result, in the following theorem, to sample from the original target distribution $\pi^X \propto \exp(-f)$ by choosing a proper regularization constant $\mu$.

**Theorem C.6** *Let $\pi^X \propto \exp(-f(x))$ be the target distribution where $f$ is convex and satisfies (1). Let $x^0 \in \mathbb{R}^d$ and $\varepsilon > 0$ be given and set*

$$\mu = \frac{\varepsilon}{\sqrt{2}\left(\sqrt{\mathcal{M}_4} + \|x^0 - x_{min}\|^2\right)} \tag{36}$$

*where $\mathcal{M}_4 = \int_{\mathbb{R}^d} \|x - x_{min}\|^4 d\pi^X(x)$ and $x_{min} \in \text{Argmin}\left\{f(x) : x \in \mathbb{R}^d\right\}$. With initial distribution $\rho_0^X$ and stepsize $\eta \asymp 1/\left(\sum_{i=1}^n L_{\alpha_i}^{\frac{2}{\alpha_i+1}} d\right)$, Algorithm 1 using Algorithm 2 as an RGO for step 1, applied to $\nu \propto \exp(-\hat{f}) = \exp(-f - \mu\|\cdot -x^0\|^2/2)$ has the iteration-complexity bound*

$$\tilde{\mathcal{O}}\left(\frac{\sum_{i=1}^n L_{\alpha_i}^{\frac{2}{\alpha_i+1}} d\left(\sqrt{\mathcal{M}_4} + \|x^0 - x_{min}\|^2\right)}{\varepsilon}\right) \tag{37}$$

*to achieve $\varepsilon$ error to $\pi^X$ in terms of total variation.*

**Proof**: Let $\rho^X$ denote the distribution of the samples generated by Algorithm 1 using Algorithm 2 as an RGO. Following the proof of Corollary 4.1 of Chatterji et al. (2020), we obtain

$$\|\rho^X - \pi^X\|_{\text{TV}} \le \|\rho^X - \nu\|_{\text{TV}} + \|\nu - \pi^X\|_{\text{TV}}$$

and

$$\|\pi^X - \nu\|_{\mathrm{TV}} \leq \frac{1}{2}\left(\int_{\mathbb{R}^d}[f(x) - \hat{f}(x)]^2 \mathrm{d}\pi^X(x)\right)^{1/2} = \frac{1}{2}\left(\int_{\mathbb{R}^d}\left(\frac{\mu}{2}\|x - x^0\|^2\right)^2 \mathrm{d}\pi^X(x)\right)^{1/2}$$

$$\leq \frac{\mu}{2}\left(\int_{\mathbb{R}^d}\left(\|x - x_{\min}\|^2 + \|x_{\min} - x^0\|^2\right)^2 \mathrm{d}\pi^X(x)\right)^{1/2}$$

$$\leq \frac{\mu}{2}\left(\int_{\mathbb{R}^d}\left(2\|x - x_{\min}\|^4 + 2\|x_{\min} - x^0\|^4\right) \mathrm{d}\pi^X(x)\right)^{1/2}$$

$$= \frac{\sqrt{2}\mu}{2}\left(\mathcal{M}_4 + \|x_{\min} - x^0\|^4\right)^{1/2} \leq \frac{\sqrt{2}\mu}{2}\left(\sqrt{\mathcal{M}_4} + \|x_0 - x_{\min}\|^2\right) = \frac{\varepsilon}{2}$$

where the last identity is due to the definition of $\mu$ in (36). Hence, it suffices to derive the iteration-complexity bound for Algorithm 1 to achieve $\|\rho^X - \pi^X\|_{\mathrm{TV}} \leq \varepsilon/2$, which is (37) in view of Proposition C.5 with $\mu$ as in (36). Note that even though the complexity result in Proposition C.5 is with respect to KL divergence, one can get the same order of complexity with respect to total variation using Pinsker inequality. ∎

## D   Solving the optimization problem

In this section, we use Nesterov's acceleration to establish the iteration-complexity for solving a general optimization problem that is nearly strongly convex and nearly smooth. This general result is then applied in Section 3 to find an approximate stationary point of $f_y^\eta$ (see Proposition 3.2).

We consider the optimization problem $\min\{g(x) : x \in \mathbb{R}^d\}$, where $g$ satisfies

$$\frac{\mu}{2}\|u - v\|^2 - \theta \leq g(u) - g(v) - \langle g'(v), u - v\rangle \leq \frac{L}{2}\|u - v\|^2 + \theta, \quad \forall u, v \in \mathbb{R}^d, \tag{38}$$

for some given $\theta > 0$, $\mu \geq 0$, and $L \geq 0$. We use Nesterov's accelerated gradient method to find a $\rho$-approximate stationary point $w$ such that $\|g'(w)\| \leq \rho$. We also establish the iteration-complexity to obtain a $\rho$-approximate stationary point.

---

**Algorithm 3** Accelerated Gradient Method

0. Let an initial point $x_0$, parameters $L, \mu > 0$ be given, and set $y_0 = x_0$, $A_0 = 0$, $\tau_0 = 1$, and $k = 0$;
1. Compute

$$a_k = \frac{\tau_k + \sqrt{\tau_k^2 + 4\tau_k L A_k}}{2L}, \quad A_{k+1} = A_k + a_k, \tag{39}$$

$$\tau_{k+1} = \tau_k + a_k \mu, \quad \tilde{x}_k = \frac{A_k y_k + a_k x_k}{A_{k+1}}; \tag{40}$$

2. Compute

$$y_{k+1} := \underset{u \in \mathbb{R}^d}{\operatorname{argmin}}\left\{\gamma_k(u) + \frac{L}{2}\|u - \tilde{x}_k\|^2\right\}, \tag{41}$$

$$x_{k+1} := \underset{u \in \mathbb{R}^d}{\operatorname{argmin}}\left\{a_k \gamma_k(u) + \frac{\tau_k}{2}\|u - x_k\|^2\right\}, \tag{42}$$

where

$$\gamma_k(u) := g(\tilde{x}_k) + \langle g'(\tilde{x}_k), u - \tilde{x}_k\rangle + \frac{\mu}{2}\|u - \tilde{x}_k\|^2; \tag{43}$$

3. Set $k \leftarrow k + 1$ and go to step 1.

---

**Lemma D.1** *For every $k \geq 0$ and $u \in \mathbb{R}^d$, we have*

$$-\theta \leq g(u) - \gamma_k(u) \leq \frac{L - \mu}{2}\|u - \tilde{x}_k\|^2 + \theta. \tag{44}$$

**Proof**: This lemma directly follows from (38) and the definition of $\gamma_k$ in (43). ∎

**Lemma D.2** *For every $k \geq 0$, the following statements hold:*

a) $A_{k+1} = La_k^2/\tau_k$;

b) $A_{k+1} \geq \frac{1}{L}\left(1 + \frac{\sqrt{\mu}}{2\sqrt{L}}\right)^{2k}$;

c) $\sum_{i=0}^{k} A_{i+1} \geq \frac{\exp(2(k+1)(\beta-1)/\beta)-1}{L(\beta^2-1)}$ *where* $\beta = 1 + \frac{\sqrt{\mu}}{2\sqrt{L}}$.

**Proof**: a) This statement directly follows from (39).

b) It is easy to see from (39), $A_0 = 0$ and $\tau_0 = 1$ that $A_1 = 1/L$. Using the definitions of $A_k$ and $\tau_k$ in (39) and (40), respectively, and the facts that $A_0 = 0$ and $\tau_0 = 1$, we easily derive that

$$\tau_k = \tau_0 + A_k\mu = 1 + A_k\mu. \tag{45}$$

It follows from (39) that

$$A_{k+1} = A_k + a_k = A_k + \frac{\tau_k + \sqrt{\tau_k^2 + 4\tau_k LA_k}}{2L} \geq A_k + \frac{\tau_k}{2L} + \frac{\sqrt{\tau_k A_k}}{\sqrt{L}} \geq \left(\sqrt{A_k} + \frac{\sqrt{\tau_k}}{2\sqrt{L}}\right)^2.$$

The above inequality and (45) imply

$$\sqrt{A_{k+1}} \geq \sqrt{A_k} + \frac{\sqrt{\tau_k}}{2\sqrt{L}} = \sqrt{A_k} + \frac{\sqrt{1 + A_k\mu}}{2\sqrt{L}} \geq \left(1 + \frac{\sqrt{\mu}}{2\sqrt{L}}\right)\sqrt{A_k}.$$

This statement now follows from the above relation and the fact that $A_1 = 1/L$.

c) Noting from b) that $A_{k+1} \geq \beta^{2k}/L$, which together with the fact $x \geq \exp((x-1)/x)$ for $x \geq 1$, implies that

$$\sum_{i=0}^{k} A_{i+1} \geq \frac{1}{L}\sum_{i=0}^{k} \beta^{2i} = \frac{\beta^{2(k+1)}-1}{L(\beta^2-1)} \geq \frac{\exp(2(k+1)(\beta-1)/\beta)-1}{L(\beta^2-1)}.$$

∎

**Lemma D.3** *For every $k \geq 0$, define*

$$t_k(u) = A_k\left[g(y_k) - g(u)\right] + \frac{\tau_k}{2}\|u - x_k\|^2, \tag{46}$$

*then for every $u \in \mathbb{R}^d$, we have*

$$\frac{\mu}{2}A_{k+1}\|y_{k+1} - \tilde{x}_k\|^2 \leq t_k(u) - t_{k+1}(u) + 2A_{k+1}\theta. \tag{47}$$

**Proof**: Using the fact $\gamma_k$ is convex and the definition of $\tilde{x}_k$ in (40), we have

$$A_k\gamma_k(y_k) + a_k\gamma_k(u) + \frac{\tau_k}{2}\|u - x_k\|^2$$

$$\geq A_{k+1}\gamma_k\left(\frac{A_ky_k + a_ku}{A_{k+1}}\right) + \frac{\tau_k A_{k+1}^2}{2a_k^2}\left\|\frac{A_ky_k + a_ku}{A_{k+1}} - \tilde{x}_k\right\|^2$$

$$= A_{k+1}\left[\gamma_k\left(\frac{A_ky_k + a_ku}{A_{k+1}}\right) + \frac{L}{2}\left\|\frac{A_ky_k + a_ku}{A_{k+1}} - \tilde{x}_k\right\|^2\right]$$

$$\geq A_{k+1}\min\left\{\gamma_k(u) + \frac{L}{2}\|u - \tilde{x}_k\|^2\right\}$$

$$= A_{k+1}\left[\gamma_k(y_{k+1}) + \frac{L}{2}\|y_{k+1} - \tilde{x}_k\|^2\right],$$

where the first identity is due to (39) and the second identity is due to the definition of $y_{k+1}$ in (41). It follows from the second inequality of (44) with $u = y_{k+1}$ and the above inequality that

$$A_{k+1}\left[g(y_{k+1}) - \theta + \frac{\mu}{2}\|y_{k+1} - \tilde{x}_k\|^2\right]$$

$$\leq A_{k+1}\left[\gamma_k(y_{k+1}) + \frac{L}{2}\|y_{k+1} - \tilde{x}_k\|^2\right]$$

$$\leq A_k\gamma_k(y_k) + a_k\gamma_k(x_{k+1}) + \frac{\tau_k}{2}\|x_{k+1} - x_k\|^2$$

$$\leq A_k\gamma_k(y_k) + a_k\gamma_k(u) + \frac{\tau_k}{2}\|u - x_k\|^2 - \frac{\tau_{k+1}}{2}\|u - x_{k+1}\|^2$$

where the last inequality is due to (42) and the fact that $a_k\gamma_k + \tau_k\| \cdot -x_k\|^2/2$ is $\tau_{k+1}$-strongly convex. Rearranging the terms in the above inequality, we obtain

$$\frac{\mu}{2}A_{k+1}\|y_{k+1} - \tilde{x}_k\|^2$$

$$\leq A_k\gamma_k(y_k) + a_k\gamma_k(u) + \frac{\tau_k}{2}\|u - x_k\|^2 - \frac{\tau_{k+1}}{2}\|u - x_{k+1}\|^2 - A_{k+1}\left[g(y_{k+1}) - \theta\right]$$

$$= A_k\left[g(y_k) - g(u)\right] + \frac{\tau_k}{2}\|u - x_k\|^2 - A_{k+1}\left[g(y_{k+1}) - g(u)\right] - \frac{\tau_{k+1}}{2}\|u - x_{k+1}\|^2$$

$$+ A_k\left[\gamma_k(y_k) - g(y_k)\right] + a_k\left[\gamma_k(u) - g(u)\right] + A_{k+1}\theta$$

$$\leq t_k(u) - t_{k+1}(u) + 2A_{k+1}\theta$$

where the identity is due to the fact that $A_{k+1} = A_k + a_k$, and the last inequality is due to (46) and the first inequality of (44). ∎

**Proposition D.4** *If $\rho \geq 2\sqrt{2}(\mu + L)\sqrt{\theta}/\sqrt{\mu}$, then the number of iterations $k_0$ to obtain a $\rho$-approximate stationary point of $g$ is at most*

$$k_0 := \frac{2\sqrt{L} + \sqrt{\mu}}{2\sqrt{\mu}} \log\left(\frac{(\mu + L)^2 d_0^2}{\rho^2}\frac{2\sqrt{L} + \sqrt{\mu}}{2\sqrt{\mu}} + 1\right). \tag{48}$$

**Proof**: It follows from the optimality condition of (41) that

$$g'(\tilde{x}_k) = (\mu + L)(\tilde{x}_k - y_{k+1}).$$

Using the above relation and summing (47) with $u = x^*$ from $k = 0$ to $k - 1$, we have

$$\frac{\mu}{2(\mu + L)^2}\sum_{i=0}^{k-1} A_{i+1}\|g'(\tilde{x}_i)\|^2 = \frac{\mu}{2}\sum_{i=0}^{k-1} A_{i+1}\|y_{i+1} - \tilde{x}_i\|^2$$

$$\leq t_0(x^*) + 2\sum_{i=0}^{k-1} A_{i+1}\theta = \frac{d_0^2}{2} + 2\sum_{i=0}^{k-1} A_{i+1}\theta,$$

where the last identity follows from the facts that $A_0 = 0$ and $\tau_0 = 1$. The above inequality and the assumption on $\rho$ imply that

$$\min_{0 \leq i \leq k-1}\|g'(\tilde{x}_i)\|^2 \leq \frac{(\mu + L)^2}{\mu}\left(\frac{d_0^2}{\sum_{i=0}^{k-1} A_{i+1}} + 4\theta\right) \leq \frac{(\mu + L)^2}{\mu}\frac{d_0^2}{\sum_{i=0}^{k-1} A_{i+1}} + \frac{\rho^2}{2}.$$

In order to show $\min_{0 \leq i \leq k-1}\|g'(\tilde{x}_i)\| \leq \rho$, it suffices to show

$$\frac{(\mu + L)^2}{\mu}\frac{d_0^2}{\sum_{i=0}^{k-1} A_{i+1}} \leq \frac{\rho^2}{2}. \tag{49}$$

Using Lemma D.2 (c) and the fact that $k \geq k_0$ where $k_0$ is as in (48), we have

$$\sum_{i=0}^{k-1} A_{i+1} \geq \frac{\exp(2k(\beta - 1)/\beta) - 1}{L(\beta^2 - 1)} \geq \frac{2(\mu + L)^2 d_0^2}{\mu\rho^2},$$

and hence (49) is proved. ∎

# E   Approximate implementations of ASF

The implementation of RGO in Algorithm 2 is exact, hence the samples of both RGO and ASF are unbiased. In contrast, it is shown in Theorem 1 of Wibisono (2019) (resp., Theorem 2 of Vempala & Wibisono (2019)) that the proximal Langevin algorithm (PLA) (resp., LMC) is biased. From the perspective of proximal sampling, we give an explanation of this fact by showing that PLA and its equivalent method, namely, proximal Langevin Monte Carlo (PLMC) of Bernton (2018), can be viewed as an instance of ASF whose implementation of RGO is inexact.

Assume the potential function $f$ is convex and smooth. Recall that PLMC iteratively generates samples as follows: given $y_k \in \mathbb{R}^d$, then the next sample $y_{k+1}$ takes the form of $y_{k+1} = \text{prox}_{\eta f}(y_k) + \sqrt{2\eta}z$ where $z \sim \mathcal{N}(0, I)$. It is easy to verify that the following algorithm gives an equivalent form of PLMC.

---
**Algorithm 4** Proximal Langevin Monte Carlo (Bernton, 2018)

---
1. Sample $x_k \sim \exp[-\frac{1}{2\eta}\|x - \text{prox}_{\eta f}(y_k)\|^2]$
2. Sample $y_{k+1} \sim \pi^{Y|X}(y \mid x_k) \propto \exp[-\frac{1}{2\eta}\|y - x_k\|^2]$

---

Next, we show that PLMC is an approximate implementation of ASF. Similar to ASF, PLMC also alternates between steps 1 and 2. More specifically, step 2 of PLMC plays the same role as step 1 of ASF, and step 1 of PLMC can be viewed as sampling from the proposal density $\propto \exp[-h_1(x)]$ without rejection, where $h_1(x) := f_{y_k}^\eta(\text{prox}_{\eta f}(y_k)) + \frac{1}{2\eta}\|x - \text{prox}_{\eta f}(y_k)\|^2 \le f_{y_k}^\eta(x)$. Since $f_{y_k}^\eta(x)$ is the potential function of the RGO in step 2 of ASF. Hence, step 1 of PLMC is an approximate implementation of the RGO. As a result, both PLMC and PLA are approximate implementations of ASF, and thus they are biased.

Following a similar argument, we show that LMC can be viewed as an instance of ASF whose implementation of RGO is inexact. Assume $f$ in the target distribution $\pi \propto \exp(-f)$ is convex and smooth and recall that the iterative step in LMC can be described as

$$y_{k+1} = y_k - \eta\nabla f(y_k) + \sqrt{2\eta}z, \quad z \sim \mathcal{N}(0, I). \tag{50}$$

We claim that the following algorithm gives an equivalent form of LMC (50) from the proximal sampling perspective.

---
**Algorithm 5** Langevin Monte Carlo

---
1. Sample $y_k \sim \pi^{Y|X}(y \mid x_k) \propto \exp[-\frac{1}{2\eta}\|x_k - y\|^2]$
2. Sample $x_{k+1} \sim \exp[-\frac{1}{2\eta}\|x - y_k + \eta\nabla f(y_k)\|^2]$

---

Indeed, steps 1 and 2 can be equivalently written as

$$x_{k+1} = y_k - \eta\nabla f(y_k) + \sqrt{\eta}z_k, \quad z_k \sim N(0, I),$$
$$y_{k+1} = x_{k+1} + \sqrt{\eta}z_k', \quad z_k' \sim \mathcal{N}(0, I),$$

where $y_{k+1}$ is the sample from step 1 in the next iteration. Combining the above identities, we have

$$y_{k+1} = y_k - \eta\nabla f(y_k) + \sqrt{\eta}(z_k + z_k') \overset{d}{=} y_k - \eta\nabla f(y_k) + \sqrt{2\eta}z, \quad z \sim \mathcal{N}(0, I).$$

Moreover, LMC and ASF share the same step 1, and step 2 of LMC equivalently generates $x_{k+1}$ from $\exp[-h_1(x)]$ where

$$h_1(x) := f(y_k) + \langle\nabla f(y_k), x - y_k\rangle + \frac{1}{2\eta}\|x - y_k\|^2. \tag{51}$$

Using the definition of $h_1$ in (51) and the convexity of $f$, we have

$$h_1(x) \le f(x) + \frac{1}{2\eta}\|x - y_k\|^2 = f_{y_k}^\eta(x).$$

Note that $f_{y_k}^{\eta}(x)$ is the potential function of the RGO in step 2 of ASF. Hence, step 2 of LMC can be interpreted as an RGO implementation with the proposal density $\exp[-h_1(x)]$ but without rejection. As a result, LMC is an approximate implementation of ASF and thus LMC is biased.

It is worth noting that many other sampling algorithms, for example, symmetric Langevin algorithm of Wibisono (2018) and Metropolis-adjusted proximal gradient Langevin dynamics of Mou et al. (2022a), can also be shown to be approximate implementations of ASF in an analogous way.

