# OpenReview forum: "A Proximal Algorithm for Sampling"
_TMLR — Accepted by TMLR_

### Review · Reviewer_Dv6M · 2023-04-27

**Summary Of Contributions:**

The paper investigates the use of the proximal sampler for sampling from semi-smooth potentials. While the convergence of the proximal sampler has been previously studied, this paper's novelty lies in describing implementations of the restricted Gaussian oracle (RGO) in the semi-smooth case and deriving complexity guarantees. The paper provides state-of-the-art results for semi-smooth sampling.

**Audience:**

Yes

**Broader Impact Concerns:**

None.

**Claims And Evidence:**

Yes

**Requested Changes:**

- Add a remark discussing the scale invariance of the results.
- In theorem statements, clarify that the number of queries is in expectation.
- In Thm. 4.4, provide guarantees in the Rényi divergence.
- The experimental comparison with the LMC algorithm is not fair, as LMC is a low-accuracy algorithm. Include a comparison with the Metropolis-Adjusted Langevin Algorithm (MALA) instead.

**Strengths And Weaknesses:**

The paper gives state-of-the-art results (at least, at the time it was written) for semi-smooth sampling, making it a valuable contribution to the field. The implementation of the restricted Gaussian oracle (RGO) and the derived complexity guarantees are novel and insightful.

---

### Review · Reviewer_wgsP · 2023-05-15

**Summary Of Contributions:**

The Alternating Sampling Framework (ASF) is a sampling algorithm to sample $\exp(-f)$ that achieves theoretical superiority, but relies on the implementation of the so called RGO at each iteration, a sampling subproblem that needs to be solved at each iteration. The RGO can be seen as the sampling analogue of the proximity operator in optimization, therefore its implementation is of practical interest.

This paper deals with the implementation of the RGO when $f$ is a semi-smooth function. They establish SOTA complexity results in the number of (sub)gradient calls to achieve epsilon accuracy in the semi-smooth setting.

**Audience:**

Yes

**Claims And Evidence:**

Yes

**Requested Changes:**

- "Departing from the standard smooth setting, the potentials are only assumed to be weakly smooth or non-smooth" and "When n = 1 and α1 = 0, f satisfying (1) is a non-smooth function. Thus, the cases we consider cover all three cases: smooth, weakly-smooth, and non-smooth." The paper does not cover arbitrary non-smoothness, right?

- "The target distribution ν is assumed to satisfy LSI or PI, but f can be non-convex in general." The paper does not cover arbitrary non convexity, right? Since at least PI is assumed.

- "[We] develop a proximal algorithm to sample from non-convex and semi-smooth potentials." The outer loop is not among your contributions.

- I don't understand the first sentence of the proof of Prop 3.4 (bottom of page 6). Could you explain more please?

- Please add more context at the beginning sentence of Section C, about the goal of this section, and the difference with the approach in the main paper.

- Regarding my comment on the quality of writing, I give some examples here:

" This type of distribution has been studied in (Chatterji et al., 2020; Chewi et al., 2021; Nguyen et al., 2021) for MCMC sampling." Don't need to sell PI or LSI here.

"We set δ = 1 in all our experiments." Write this sentence in the exp section rather than the theory section.

Section 6 does not help the paper (which is about implementing the RGO) and is already known (I believe). Maybe, merge section 6 with section E in the appendix.


--------------------------------------------------------------------------------------------------------------
In summary, I don't have any concerns about the results in this paper (except bottom of Page 6), but I do have concerns about the clarity of the paper, for a non expert reader.

**Strengths And Weaknesses:**

**Strengths:**

- The implementation of the RGO is important in practice. First, because the outer loop (ASF) is going to be important in the future, given the recent results on this sampler. Second, because we do not have many closed form formula for the RGO but we do have **exact** subroutines (as the one presented in the paper). So, investigating these exact subroutines is of practical interest.

- The result in this paper on the RGO (the inner loop) can be combined with known results on the ASF (the outer loop) which enables to provide an end-to-end guarantee in terms of number of gradient calls to achieve epsilon accuracy.

**Weaknesses:**

- The paper is not well written. The topic of the paper is implementing the RGO. Each section should be related to this topic. The content of each section should be dedicated to the topic of the section.

-  The introduction is too confusing for such a simple situation: there is an outer loop (ASF) that is well understood but requires implementation of the RGO (inner loop). The inner loop has been studied when f is convex nonsmooth, this paper is about the inner loop when f is semi smooth, under various other assumptions: LSI, PI, log concave. The authors could simplify and shorten the introduction.
- Semi-smoothness should be motivated. I know that TMLR is mainly about correctness of the paper, but why is semi-smoothness relevant in practice? Besides, is it easy to learn the semi-smoothness parameters?

---

### Review · Reviewer_n7Lu · 2023-05-17

**Summary Of Contributions:**

The key contribution is the development and analysis of sampling from distributions with possibly non-covex and semi-smooth potencial.
The algorithm is heavily based on: a) alternating sampling framework ASF that has been dveloped for semi-smooths (but convex) case in Lee et al. 2021 and extended to distrbutions satisfying logarithmic sobilve inequlitiy LSI or Poincare inequlity PI functional inequlities by Chen at. al.; b) ideas from Liang and Chen 2022 that treats the same setting as the current paper but with convex potencial. In particular, Liang and Chen 2022 develop practical algorithm for sampling from restricted gaussian oracle which is critical for efficient implementation of ASF.



**Audience:**

Yes

**Broader Impact Concerns:**

methdological work; no ethical considerations.

**Claims And Evidence:**

Yes

**Requested Changes:**

- a detailed discussion comparing current work and Liang and Chen 2022 should be inlcuded.
- In Lemma A.3.  is f' a derivative of subdifferential? I though f is not differentiable

**Strengths And Weaknesses:**

Strength:
- The paper is well written
- th algorithm works in a semi-smooth non convex setting extending previous works that work for with semi-smooth and convex or smooths and non-convex settings
- proposed algorithm seems to be competitive by looking at Table 1.

Weaknesses:
- in view Liang and Chen 2022 this work seems incremental.

---

### Review · Reviewer_u2Fa · 2023-05-18

**Summary Of Contributions:**

This paper examines the Alternating Sampling Framework (ASF) for sampling from a target distribution proportional to $\exp(-f)$, where $f$ is a semismooth (as defined in the paper) and nonconvex function. The ASF framework consists of two steps performed at each iteration. The first step involves Gaussian sampling, while the second step requires sampling from a proximal version of $\exp(-f)$, referred to as RGO.

To implement the RGO step, the authors propose a Rejection/Acceptance sampling mechanism by bounding the potential function with a quadratic function. They determine this new function by finding a loose stationary point using Nesterov's accelerated algorithm. The paper provides guarantees that the number of rounds required for each iteration is bounded.

**Audience:**

Yes

**Broader Impact Concerns:**

Theoretical work. No broader impact concerns.

**Claims And Evidence:**

No

**Requested Changes:**

### Requested Changes

- add the reference Balasubramanian et al. 2022 (LMC in the non-convex setting), Dalalyan et al. 2019 (LMC and KLMC in non-strongly log-concave setting).
- page 2: `one and` -> `and one`
- the hyperlinks (references, equations. etc.) do not work
- page 3: Use `citep` for the last reference.
- To avoid possible confusion, it is worth mentioning that the distributions and their densities are considered to be equivalent in the paper.
- Rewrite Algorithm 4 in terms of the function $f_y^{\eta}$. In the appendix, the parameters are different and do not translate well to the sampling setting.
- page 7: `Combing` -> `Combining`
- In the last line of (12) $\times$ operator could be added to increase the clarity of the formula.
- Page 7: the equation after `which together with (7) implies that`. The term $\frac{1}{\eta} ||w||^2$ is missing on the third line.
- Use the LaTeX environment `{align}` to align  the equations.
- Page 7: `Note $\delta$ is a tunable parameter.` ->  `Note that $\delta$ is a tunable parameter.`
- Add more complicated experiments that confirm the results of the theorem. It would be useful to see that the expected number of rejections indeed has the upper-bound mentioned in the theorems for .

**Strengths And Weaknesses:**


#### Strengths

* The class of semi-smooth functions is much larger than the one of the smooth functions.
* The application Nesterov's accelerated algorithm implemented to find the majorizing density.
* The paper extends the setting by Liang & Chen.
#### Weaknesses

-   The nonconvexity arising from (1) does not cover the entire class of nonconvex functions. In particular, (1) implies a quadratic lower bound on the function $f$. This means that by adding a square norm with a sufficiently large multiplier, a convex function can be obtained.
-   The coefficients $L_i$ and $\alpha_i$ for the example of a semismooth potential displayed on page 3 are not specified.
-   In the general case, it is unclear how to find or approximate the constants $L_i$ and $\alpha_i$.
-   None of the theorems are proven in the paper. Even the theorems with seemingly "obvious" proofs should be properly written and explained.
-   The quality of writing in the paper is insufficient, lacking necessary mathematical details.

##### Experiments

-   The chosen step size for LMC in the experiments is significantly smaller (depends on the dimension) than the theoretical step size, which  is of order ($\approx 1/L$) and does not depend on the dimension.
-   The target distribution used in the experiments is overly simplistic, and the dimension of the space is very small ($d=5$).

#### Question

-   Regarding the RGO discussed in the paper, which has a broad form allowing for almost any potential function, am I correct in understanding that the proposed method solves the exact sampling problem through a finite number of rejection-acceptance steps? If that is the case, it is possible that there is a hidden dependence on the dimension, either in the multiple parameter conditions or potentially missing in the proof.

---

### Decision · Action_Editors · 2023-06-27

**Recommendation:** Accept as is

**Comment:**

The author study a certain subroutine (RGO) of the Alternating Sampling Framework (ASF). ASF is a sampling algorithm with favorable theoretical properties relying on the call of a restricted Gaussian oracle (RGO) in each iteration. RGO is a sampling subproblem that can be seen as an  analogue of the proximity operator in optimization.

In particular, this paper deals with the implementation of the RGO in the semi-smooth regime. The authors establish new SOTA complexity results (in the number of subgradient calls to achieve epsilon accuracy) in this regime.

- The paper studies an important problem: how to implement RGO in practice.
- The extension to semi-smoothness is interesting since the class of semi-smooth functions is much larger than the class of smooth functions.
- The paper is contains interesting theoretical results that appear to be correct.
- The results are of a somewhat incremental nature, but still interesting (this is an extension of the results of Liang & Chen).
- The implementation of the RGO is important in practice.
- The writing quality was criticized by several reviewers (e.g., intro is more confusing than it has to bel; semi-smoothness can be motivated in a better way).

More pros and cons were mentioned in the reviews.

In summary, this is a good paper (after the revision), and I recommend for it to be accepted.

Action Editor

**Audience:**

Yes, the article will be interesting to a community of ML researchers interested in sampling algorithms. Sampling is the backbone of several ML subroutines.

**Claims And Evidence:**

While the original submission did not provide sufficient justification for all claims, this has been corrected during the author-reviewer discussion. By consensus of the reviewers, the theoretical claims provided in the paper seem correct, and proofs were provided.